# AMPK signaling to acetyl-CoA carboxylase is required for fasting- and cold-induced appetite but not thermogenesis

Sandra Galic[1,2]*, Kim Loh[1,2], Lisa Murray-Segal[1,2], Gregory R Steinberg[3,4], Zane B Andrews[5,6,7], Bruce E Kemp[1,2,8]*

[1]Department of Medicine, University of Melbourne, Fitzroy, Australia; [2]St. Vincent's Institute of Medical Research, Melbourne, Australia; [3]Division of Endocrinology and Metabolism, Department of Medicine, McMaster University, Hamilton, Canada; [4]Department of Biochemistry and Biomedical Sciences, McMaster University, Hamilton, Canada; [5]Biomedicine Discovery Institute, Faculty of Medicine, Nursing and Health Sciences, Monash University, Clayton, Australia; [6]Department of Physiology, Monash University, Clayton, Australia; [7]Monash Biomedicine Discovery Institute, Monash University, Clayton, Australia; [8]Mary MacKillop Institute for Health Research, Australian Catholic University, Fitzroy, Australia

**Abstract** AMP-activated protein kinase (AMPK) is a known regulator of whole-body energy homeostasis, but the downstream AMPK substrates mediating these effects are not entirely clear. AMPK inhibits fatty acid synthesis and promotes fatty acid oxidation by phosphorylation of acetyl-CoA carboxylase (ACC) 1 at $Ser^{79}$ and ACC2 at $Ser^{212}$. Using mice with $Ser^{79}Ala/Ser^{212}Ala$ knock-in mutations (ACC DKI) we find that inhibition of ACC phosphorylation leads to reduced appetite in response to fasting or cold exposure. At sub-thermoneutral temperatures, ACC DKI mice maintain normal energy expenditure and thermogenesis, but fail to increase appetite and lose weight. We demonstrate that the ACC DKI phenotype can be mimicked in wild type mice using a ghrelin receptor antagonist and that ACC DKI mice have impaired orexigenic responses to ghrelin, indicating ACC DKI mice have a ghrelin signaling defect. These data suggest that therapeutic strategies aimed at inhibiting ACC phosphorylation may suppress appetite following metabolic stress.
DOI: https://doi.org/10.7554/eLife.32656.001

*For correspondence:
sgalic@svi.edu.au (SG);
bkemp@svi.edu.au (BEK)

**Competing interests:** The authors declare that no competing interests exist.

## Introduction

There are 1.9 billion adults overweight or obese worldwide, placing them at increased risk of developing type two diabetes, cardiovascular disease, chronic kidney disease and cancer (World Health Organization, Fact Sheet N°311, *Obesity and overweight*). Weight gain develops as a consequence of an imbalance between the individual's energy intake and energy expenditure, as occurs with over nutrition and sedentary life styles. Current weight loss strategies are typically aimed at generating a negative energy balance through calorie restriction (dieting) or increases in energy expenditure (exercise). In recent years brown fat thermogenesis has emerged as a potential alternative to exercise to increase energy expenditure in humans and has led to increased interest in the development of pharmacological activators of thermogenesis (*Cypess and Kahn, 2010*; *Whittle et al., 2011*). However, weight loss strategies aimed at prolonged negative energy balance typically fail due to compensatory increases in appetite (*Doucet et al., 2000*; *Sumithran et al., 2011*). The underlying

molecular mechanisms by which increases in energy demand are coupled to increased caloric intake are not fully understood. Thus, gaining a better understanding of these mechanisms may reveal new therapeutic strategies for the treatment of obesity.

The serine/threonine kinase AMP-activated protein kinase (AMPK) is a well-recognized regulator of whole-body energy balance. AMPK is activated by intracellular energy depletion induced by muscle contraction, nutrient deprivation and hypoxia and acts to restore ATP levels by inhibiting anabolic energy-consuming and promoting catabolic ATP-generating pathways. Factors that lead to activation of AMPK increase appetite by promoting orexigenic neuropeptide expression in the hypothalamus, including agouti-related transcript (AgRP), neuropeptide Y (NPY), orexins and melanin-concentrating hormone (MCH), while suppressing anorexigenic signals, such as proopiomelanocortin (POMC) and cocaine and amphetamine-regulated transcript (CART) (*Claret et al., 2007*). AMPK activation simultaneously reduces energy expenditure by suppressing sympathetic outflow to brown adipose tissue and reducing heat dissipation associated with thermogenesis, with the final goal to achieve energy balance (*López et al., 2010*; *Martínez de Morentin et al., 2014*). However, the AMPK substrates involved in these specific responses remain unclear, due to AMPK's pleiotropic effects on metabolic pathways and cross-talk with other energy sensors, including Sirt1 and mTOR.

One of the most widely reported AMPK functions is the regulation of lipid metabolism through phosphorylation of acetyl-CoA carboxylase 1 (ACC1) at $Ser^{79}$ and ACC2 at $Ser^{212}$. ACC1 $Ser^{79}$/ACC2 $Ser^{212}$ phosphorylation inhibits the production of malonyl-CoA, a substrate for fatty acid synthase (FAS) and precursor for the de novo synthesis of palmitate. In addition, malonyl-CoA is a potent inhibitor of the mitochondrial carnitine palmitoyl transferase 1 (CPT1) and limits fatty acids β-oxidation. Inhibition of ACC activity by AMPK has previously been proposed as an essential step in the regulation of the appetite and thermogenesis (*Andrews et al., 2008*; *Gao et al., 2007*; *López et al., 2008*, *2010*; *Martínez de Morentin et al., 2014*). We generated a mouse line with alanine knock-in mutations of ACC1 $Ser^{79}$ and ACC2 $Ser^{212}$ (ACC DKI mice) that render ACC activity and malonyl-CoA production insensitive to AMPK. We report that ACC1 $Ser^{79}$/ACC2 $Ser^{212}$ phosphorylation is selectively important for promoting food intake and determining fuel utilization under cold stress, while AMPK-regulated energy expenditure and capacity for thermogenesis are independent of AMPK-ACC signaling. Furthermore, our results show that inhibition of ACC1 $Ser^{79}$/ACC2 $Ser^{212}$ phosphorylation leads to ghrelin insensitivity and indicate that increased food intake in response to metabolic stress requires intact ghrelin receptor-mediated activation of the AMPK-ACC pathway.

## Results

### Inhibition of ACC1 $Ser^{79}$/ACC2 $Ser^{212}$ phosphorylation increases adipose tissue lipid synthesis capacity, but does not promote adiposity

Changes in lipid metabolism in adipose tissue are known to have profound effects on whole body energy homeostasis (*Lee et al., 2015*; *Lodhi et al., 2012*; *Vernochet et al., 2012*). To gain insight into the importance of ACC1 $Ser^{79}$/ACC2 $Ser^{212}$ phosphorylation for body mass regulation, we assessed ACC enzyme activity and lipid metabolism in adipose tissue of wild-type and ACC DKI mice. Activity of ACC1, the main ACC isoform in murine adipose tissue, was increased in brown and subcutaneous inguinal white fat of ACC DKI mice (*Figure 1A*) and correlated with increased lipogenesis in vivo (*Figure 1B*), suggesting an increased propensity for lipid accumulation in ACC DKI adipose tissue. In contrast, inhibition of ACC $Ser^{79}$/ACC2 $Ser^{212}$ phosphorylation did not affect the capacity for $^{14}$C-palmitate oxidation in brown or subcutaneous fat (*Figure 1C*). To assess the impact of increased adipose tissue lipogenesis on body mass regulation, we monitored weekly body weights of male mice on standard chow diet and housed at room temperature (18–20°) from 3 to 30 weeks of age (*Figure 1D*). We found that ACC DKI mice tended to be leaner than age-matched wild-type mice from 15 weeks of age (*Figure 1D*) with a small reduction in total adiposity at older age as determined by nuclear magnetic resonance (*Figure 1E*). Furthermore, measurements of individual fat pad weights indicated that the reduced adiposity was due to changes in white fat mass, as brown fat mass was similar between genotypes (*Figure 1F*). These data suggest the presence of a mechanism, by which loss of ACC1 $Ser^{79}$/ACC2 $Ser^{212}$ phosphorylation may confer an overall

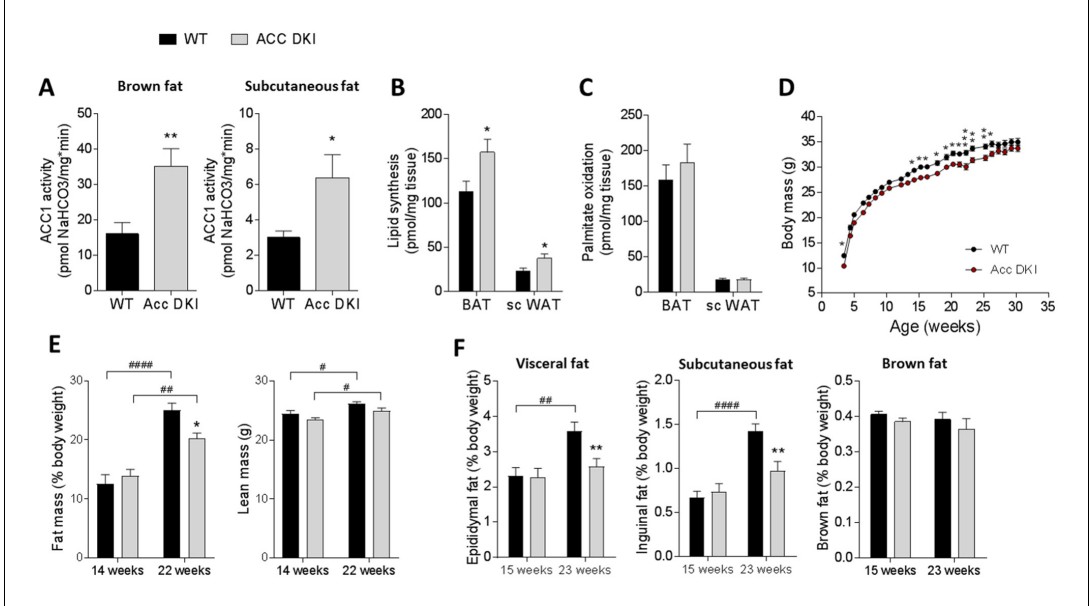

**Figure 1.** Inhibition of ACC1 Ser[79]ACC2 Ser[212] phosphorylation increases adipose tissue lipid synthesis, but does not promote adiposity. (**A**) ACC1 activity in brown fat and subcutaneous fat of 12–15 week old male mice (n = 9). (**B**) Incorporation of [³H]acetate into the total tissue lipid fraction in brown fat and subcutaneous fat as a measure of de novo lipogenesis (n = 6). (**C**) Palmitate oxidation in brown and subcutaneous fat explants ex vivo (n = 13–15). (**D**) Weight curves of male wild-type and ACC DKI mice on a chow diet (n = 14). (**E**) Assessment of body composition by NMR scanning showing percentage of fat mass and total lean mass (n = 7–10). (**F**) Epididymal, inguinal subcutaneous and brown fat pad weights expressed as percentage of total body weight in male mice at indicated ages (n = 8–15). All data were derived from mice housed at an environmental temperature of 18–20°C. Data are expressed as means ± s.e.m. (**A**)-(**C**) *p<0.05, **p<0.01 represent differences between genotypes in the specified tissue type as determined by unpaired t-test, two-tailed. (**D**) *p<0.05, **p<0.01, ***p<0.001 genotype differences at a given time point determined by 2-way repeated measures ANOVA with Bonferroni post-hoc test. (**E**)-(**F**) **p<0.01 differences between genotypes within an age group; p<0.05, p<0.01, p<0.0001 differences between age groups for a given genotype as determined by 2-way ANOVA with Bonferroni post hoc test.

DOI: https://doi.org/10.7554/eLife.32656.002

The following source data is available for figure 1:

**Source data 1.** Sample size, mean and s.e.m. and statistical calculations are presented.
DOI: https://doi.org/10.7554/eLife.32656.003

negative energy balance leading to modest reductions in adiposity over time, despite promoting increased capacity for lipid synthesis in isolated tissue.

## ACC DKI mice fail to increase food intake in response to cold exposure

To investigate whether the negative energy balance of ACC DKI mice could be attributed to changes in energy expenditure, in our subsequent experiments we used mice at an average of 12 weeks of age, when body mass and total fat and lean mass were similar between genotypes (*Figure 1D and E*). We housed mice individually and measured metabolic parameters continuously for the duration of 72 hr using indirect calorimetry. Thermogenesis contributes significantly to overall metabolic rate at temperatures below thermoneutrality (8% increase in energy expenditure per 1°C drop in temperature below 28°C (*Virtue et al., 2012*) and AMPK has previously been implicated in regulating the capacity of brown fat for heat generation (*López et al., 2010*; *Martínez de Morentin et al., 2014*; *Martínez de Morentin et al., 2012*; *Mottillo et al., 2016*; *Whittle et al., 2012*). Therefore, in addition to collecting measurements at room temperature (21°C), we also investigated the metabolic profile of mice exposed to cold (14°C). Separate groups of mice were housed under thermoneutral conditions (28°C), when the proportion of energy expended for active heat production is minimal.

We found that the total energy expended was similar between wild-type and ACC DKI mice regardless of environmental temperature (*Figure 2A–C*). Consistent with this, plasma levels of hormones known to be important activators of thermogenesis, such as thyroxine, epinephrine and

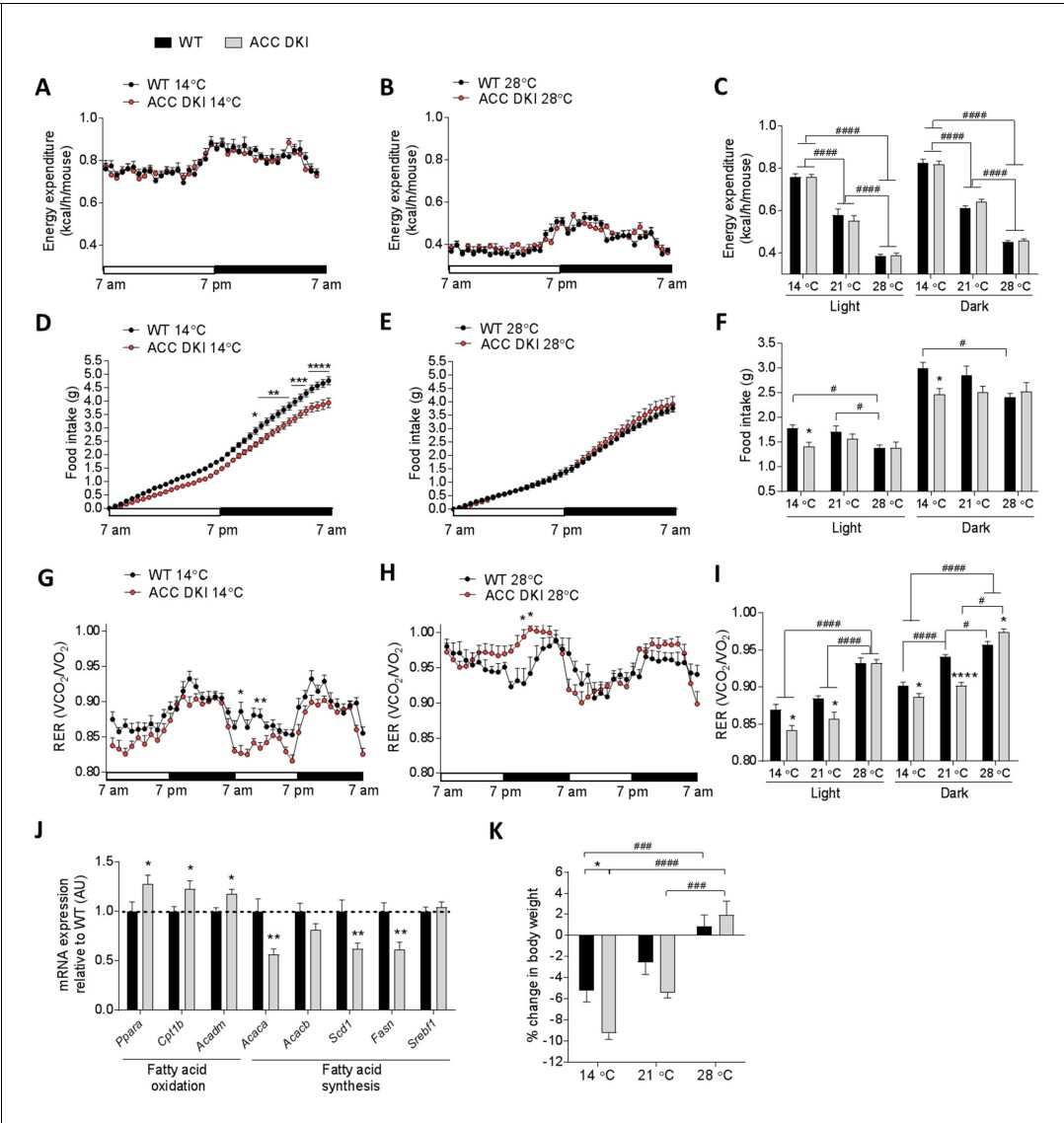

**Figure 2.** Metabolic parameters of wild-type and ACC DKI male mice exposed to various temperatures. (A–C) Energy expenditure, (D–F) cumulative food intake and (G–I) respiratory exchange ratio (RER) in 10–14 week old, ad libitum fed, wild-type and ACC DKI male mice after 72 hr of exposure to indicated temperatures (n = 8). (A, B, D, E, G, H) Hourly averages of metabolic parameters of mice exposed to 14°C and 28°C. Black horizontal bars represent the dark period in a 12 hr light/dark cycle (7am-7pm). *p<0.05, **p<0.01, ***p<0.001, ****p<0.0001 represent differences between genotypes for a given time point as determined by 2-way repeated measures ANOVA with Bonferroni post-hoc test. (C, F, I) 12 hr averages of at least two consecutive day and night cycles of metabolic parameters of mice exposed to indicated temperatures. *p<0.05, ****p<0.0001 differences between genotypes within a temperature group and; # p<0.05, p<0.0001 differences between parameters for a given genotype at different temperatures as determined by 2-way ANOVA with Bonferroni post-hoc test. (J) mRNA expression profile in brown fat of mice exposed to 14°C for 72 hr (n = 12). *p<0.05, **p<0.01 genotype difference in mRNA expression for a given gene as determined by unpaired t-test, two-tailed. (K) Percentage body weight change of wild-type and ACC DKI mice exposed to various temperatures for 72 hr. *p<0.05 represents differences between genotypes and p<0.001, p<0.0001 represent differences in body weight loss at different temperatures as determined by 2-way ANOVA and Bonferroni post-hoc test. All data are presented as mean ± s.e.m.

DOI: https://doi.org/10.7554/eLife.32656.004

The following source data and figure supplements are available for figure 2:

**Source data 1.** Sample size, mean and s.e.m. and statistical calculations are presented.
DOI: https://doi.org/10.7554/eLife.32656.010

**Figure supplement 1.** Inhibition of ACC Ser[79]/Ser[212] phosphorylation has no effect on thermogenic capacity.
DOI: https://doi.org/10.7554/eLife.32656.005

**Figure supplement 1—source data 1.** Sample size, mean and s.e.m.

*Figure 2 continued on next page*

*Figure 2 continued*

DOI: https://doi.org/10.7554/eLife.32656.006

**Figure supplement 1—source data 2.** Western blots are presented for *Figure 2—figure supplement 1*.

DOI: https://doi.org/10.7554/eLife.32656.007

**Figure supplement 2.** Plasma lipid concentrations and ambulatory activity of WT and ACC DKI mice exposed to various temperatures.

DOI: https://doi.org/10.7554/eLife.32656.008

**Figure supplement 2—source data 1.** Sample size, mean and s.e.m.

DOI: https://doi.org/10.7554/eLife.32656.009

norepinephrine and were not different between wild-type and ACC DKI mice exposed to 14°C or 28°C (*Figure 2—figure supplement 1, A–C*)). Furthermore, protein (*Figure 2—figure supplement 1D*) and mRNA expression (*Figure 2—figure supplement 1E*) of UCP1 and other thermogenic markers in brown fat were not affected by the DKI mutation, showing that inhibition of ACC1 Ser[79]/ACC2 Ser[212] phosphorylation is of no consequence for brown fat thermogenic capacity.

We next examined the cumulative food intake of mice housed in metabolic chambers at 14°C, 21°C and 28°C (*Figure 2*, D-F). At sub-thermoneutral temperatures wild-type mice increased their food intake to meet the energy demands of thermogenesis required to maintain body temperature. In contrast, ACC DKI mice showed no increases in appetite and, regardless of the degree of cold exposure, consumed the same amount of food as if housed at thermoneutrality. Given the sharp increase in energy expenditure required for heat generation at lower temperatures, ACC DKI mice would be expected to suffer an energy imbalance that becomes greater depending on the degree of cold exposure.

The increased metabolic rate at cold exposure is supported by an increase in substrate mobilization, with brown fat being largely responsible for determining the substrate utilization profile (*Bartelt et al., 2011*; *Stanford et al., 2013*). ACC DKI mice showed reduced respiratory exchange ratio (RER) at temperatures below 28°C (*Figure 2G-I*), indicating that in response to the reduced energy intake, ACC DKI mice oxidized fat at the expense of carbohydrate. Furthermore, measurement of the expression profile of metabolic genes in brown fat of mice housed at 14°C for 72 hr revealed a significant upregulation of genes involved in fatty acid β-oxidation and a concomitant reduction of lipogenic genes (*Figure 2J*), suggesting that brown fat of ACC DKI mice has undergone specific metabolic adaptations for greater utilization of fatty acids after cold stress. This difference in RER appeared to be a specific response to the cold exposure, as ACC DKI mice housed at thermoneutral conditions showed either no difference when compared to wild-type littermates or a reversal to preferential utilization of carbohydrates (*Figure 2H and I*). Consistent with reduced oxidation of lipids, ACC DKI mice had significantly elevated serum triglyceride levels when housed at thermoneutrality (*Figure 2—figure supplement 2A*), while an acute exposure to 30°C also revealed increased plasma non-esterified fatty acid (NEFA) concentrations (*Figure 2—figure supplement 2B*). Together with the increased capacity for fatty acid synthesis (*Figure 1B*), this indicates that in the absence of metabolic stress, inhibition of ACC1 Ser[79]/ACC2 Ser[212] phosphorylation may indeed predispose to increased adiposity.

ACC DKI mice also showed a strong tendency for reduced ambulatory activity (*Figure 2—figure supplement 2, C–E*), particularly at 14°C, which may have been a consequence of reduced food-seeking behavior (*Sakkou et al., 2007*) associated with the reduced appetite in ACC DKI mice. However, at sub-thermoneutral temperatures, changes in activity levels have been shown to contribute little to total daily energy expenditure as most of the energy expended is due to brown fat thermogenesis (*Virtue et al., 2012*). Consistent with this, any compensatory reductions in activity levels were insufficient to impact on overall energy expenditure (*Figure 2A–C*) and prevent body weight loss in ACC DKI mice exposed to cold (*Figure 2K*).

## ACC DKI mice have reduced food intake in response to metabolic stress

AMPK activation in the hypothalamus has previously been shown to stimulate appetite in response to various hormones and nutritional states (*Andersson et al., 2004*; *Andrews et al., 2008*; *López et al., 2008*; *Minokoshi et al., 2004*) and hypothalamic ACC1 Ser[79]/ACC Ser[212]

phosphorylation has been shown to increase in response to both fasting (*López et al., 2008*) and cold exposure (*Roman et al., 2005*).

To investigate the effect of the ACC knock-in mutation in the hypothalamus, we performed immunoblotting using whole hypothalamus tissue from wild-type and ACC DKI mice housed at standard animal house temperatures and detected no compensatory change in AMPK Thr172 phosphorylation or ACC protein abundance (*Figure 3—figure supplement 1A*). Furthermore, enzyme activity of ACC1, the only brain ACC isoform detectable by immunoblotting (*Figure 3—figure supplement 1B*), was increased in whole hypothalamus tissue isolated from overnight fasted ACC DKI mice (*Figure 3A*).

We next analyzed immunoreactivity of the immediate early gene transcription factor c-Fos as a readout for neuronal activity using brain sections from mice acutely exposed to either 4°C or 30°C for 90 min (*Figure 3B*). While the short-term cold exposure increased c-Fos staining in multiple nuclei across the hypothalamus, we could not detect any genotype differences in neuronal activation in hypothalamic areas, known to be involved in the regulation of thermogenesis and energy expenditure, such as the dorsomedial hypothalamus (DMH), paraventricular nucleus (PVN) or ventromedial hypothalamus (VMH) (*Figure 3—figure supplement 2, A,B*). However, we found the most apparent reduction of c-Fos-positive cells specifically in the arcuate nucleus (ARC) of ACC DKI mice at 4°C (*Figure 3B*). The ARC of the hypothalamus is considered the primary nutrient-sensing center regulating appetite by responding to nutritional and hormonal cues from the periphery. To investigate whether ACC DKI mice may have altered peripheral signals following temperature stress, we measured plasma concentrations of metabolic hormones of mice housed at cold stress or thermoneutrality for 72 hr (*Figure 3C*). Exposure of mice to either 14°C or 28°C did not change overall plasma leptin concentrations and the amounts were similar in wild-type and ACC DKI mice with both conditions. In contrast, plasma insulin and ghrelin concentrations responded to temperature stress in a reciprocal manner. While there was no genotype difference in the plasma levels of insulin, the increase in plasma ghrelin was exacerbated in cold-exposed ACC DKI mice, suggesting a potential defect in the response to the ghrelin signal in these mice.

Ghrelin is an orexigenic hormone secreted from the endocrine cells of the stomach that has been well described during the fasting-feeding transition, whereby the plasma ghrelin concentration increases with fasting and decrease immediately after a meal (*Cummings et al., 2001*; *Tschöp et al., 2001*). Ghrelin has previously been reported to increase food intake through AMPK activation in the hypothalamus (*Andrews et al., 2008*; *López et al., 2008*). To further explore the possibility that ghrelin action may be affected in ACC DKI mice, we investigated the appetite response of mice after an overnight fast, when ghrelin signaling would be expected to be most relevant. We found that ACC DKI mice consumed significantly less food after a 5 hr period of refeeding (*Figure 3D*). This correlated with decreased hypothalamic mRNA expression of the orexigenic neuropeptides *Npy* and *Agrp*, whereas expression of the anorexigenic *Pomc* and *Cartpt* was similar between genotypes (*Figure 3E*). In contrast, we did not detect any difference in the weekly *ad libitum* food intake (*Figure 3F*), suggesting that, similar to our previous observations during cold exposure, the effects of the knock in mutation on appetite only become apparent after a metabolic challenge, such as an overnight fast.

To gain insight into the potential endocrine signals responsible for the appetite difference after fasting, we measured plasma concentrations of ghrelin and the anorexigenic leptin and insulin, all of which have previously been suggested to regulate AMPK activity and food intake during fasting and refeeding (*López et al., 2008*; *Minokoshi et al., 2004*). Leptin and insulin concentrations decreased after overnight fasting without significant differences between genotypes at either fasted or *ad libitum* fed conditions (*Figure 3G*). In contrast, plasma ghrelin was significantly increased in fasted ACC DKI mice, comparable to the increase observed during cold exposure (*Figure 3G*). Together these results show that in addition to increasing energy intake during cold exposure, AMPK-ACC signaling also contributes to the orexigenic response to fasting, as another form of metabolic stress. Both conditions are accompanied by a rise in plasma ghrelin levels, which is exacerbated in ACC DKI mice, suggesting that ghrelin signaling may be the pathway responsible for the reduced appetite response in these mice.

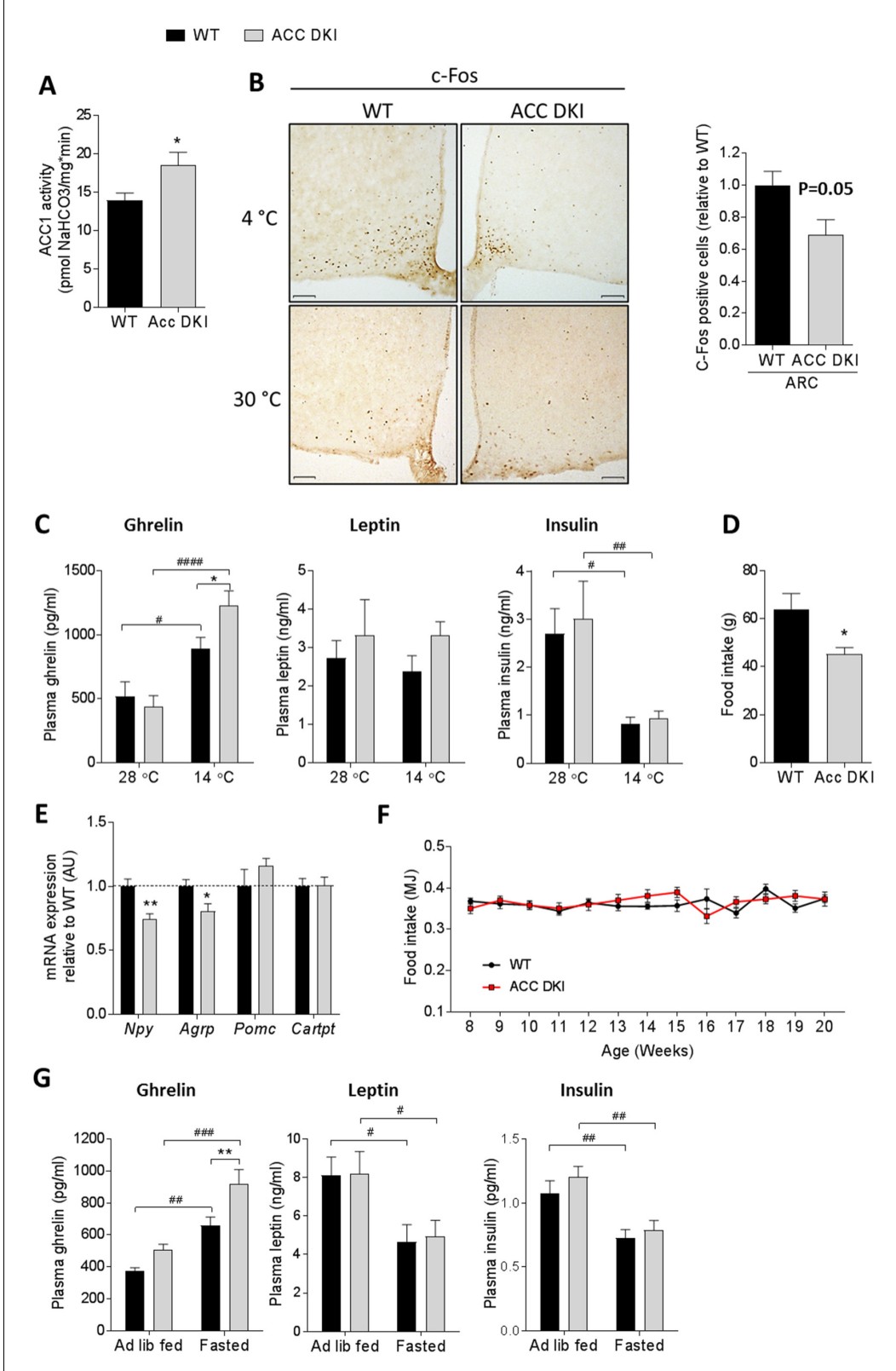

**Figure 3.** ACC DKI mice have reduced food intake in response to metabolic stress. (A) ACC1 enzyme activity in hypothalamus of fasted mice (n = 6). (B) c-Fos immunohistochemistry in hypothalamus cross sections of mice acutely (90 min) exposed to indicated temperatures. Images are showing c-Fos staining in the arcuate nucleus. Bar graph represents quantification of c-Fos immunoreactivity in the arcuate nucleus, expressed as number of c-Fos positive cells within a defined region relative to wild-type (n = 4 mice/group). Scale bars = 100 µm. (C) Plasma hormone concentrations after exposure

*Figure 3 continued on next page*

*Figure 3 continued*

to indicated temperatures for 72 hr (n = 11–12 for ghrelin, n = 8 for insulin and leptin). (D) Food intake after overnight fast and 5 hr refeed (n = 8). (E). mRNA expression of neuropeptides in the hypothalamus of overnight fasted mice (n = 8). (F) Weekly ad libitum food intake in male mice from 8 to 20 weeks of age (n = 8–10). (G) Plasma hormone concentrations in ad libitum fed and overnight fasted mice (n = 15-29 for ghrelin, n = 8 for leptin, n = 12–13 for insulin). Data are expressed as means ± s.e.m. (A), (B), (D) and (E) were analyzed by unpaired t-test, two-tailed with *p<0.05, **p<0.01 representing differences between genotypes. (C) and (G) were analyzed by 2-way ANOVA with Bonferroni post-hoc test with *p<0.05 representing differences between genotypes at a given temperature and p<0.05, p<0.01, p<0.0001 are differences between hormone levels at different temperatures (C) or feeding states (G) for a given genotype as determined by 2-way ANOVA with Bonferroni post-hoc test. (F) Data were analyzed by 2-way repeated measures ANOVA and Bonferroni post-hoc test. All data are presented as mean ± s.e.m.

DOI: https://doi.org/10.7554/eLife.32656.011

The following source data and figure supplements are available for figure 3:

**Source data 1.** Sample size, mean and s.e.m. and statistical calculations are presented.
DOI: https://doi.org/10.7554/eLife.32656.016
**Figure supplement 1.** Expression and phosphorylation of ACC in mouse hypothalamus tissue.
DOI: https://doi.org/10.7554/eLife.32656.012
**Figure supplement 1—source data 1.** Western blots are presented for *Figure 3—figure supplement 1*.
DOI: https://doi.org/10.7554/eLife.32656.013
**Figure supplement 2.** c-Fos immunohistochemistry in hypothalamus cross sections of mice acutely exposed to 4°C.
DOI: https://doi.org/10.7554/eLife.32656.014
**Figure supplement 2—source data 1.** Sample size, mean and s.e.m.
DOI: https://doi.org/10.7554/eLife.32656.015

## ACC DKI mice show reduced sensitivity to the orexigenic actions of exogenous ghrelin

To assess ghrelin sensitivity of ACC DKI mice, we subsequently analyzed food intake and neuropeptide expression following exogenous ghrelin administration. Using an intraperitoneal injection with 1 µg/g acyl-ghrelin we detected a 3.3-fold increase in plasma ghrelin concentrations at 1 hr after ghrelin treatment (*Figure 4—figure supplement 1A*). Cumulative food intake 2 hr after intraperitoneal ghrelin injection was significantly blunted in ACC DKI mice (*Figure 4A*) consistent with AMPK-ACC signaling being a component of the ghrelin pathway. Enzyme activity assays in whole hypothalamus tissue isolated from mice 1 hr after intraperitoneal injection showed that ghrelin was unable to reduce ACC1 activity of ACC DKI mice (*Figure 4B*) and that this effect was not due to defects in AMPK activation (*Figure 4C*). Consistent with this, mRNA expression of the orexigenic neuropeptides *Npy* and *Agrp* were lower in the hypothalamus of ghrelin-injected ACC DKI mice, while *Pomc* and *Cartpt* were unchanged (*Figure 4D*), similar to the expression pattern of overnight fasted mice (*Figure 3E*).

In addition to its well-known acute effects on food intake, ghrelin also promotes adiposity chronically independent of hyperphagia. These effects are thought to be associated with reduced expression of *Ucp1* in brown fat and increased expression of lipogenic genes in white adipose tissue (*Theander-Carrillo et al., 2006*). To investigate whether AMPK phosphorylation of ACC is also required for ghrelin's chronic effects, we administered ghrelin (30 µg/day/mouse) to wild-type and ACC DKI mice over a period of 14 days using osmotic mini-pumps as previously described (*Andrews et al., 2008*). Mice were housed at room temperature during the experiment. We found that after one week of ghrelin treatment, wild-type mice started to gain weight and reached a maximum of up to 4% increase in body mass when compared to saline-treated controls at day 12 of treatment (*Figure 4E*). In contrast, ACC DKI mice did not show any significant changes in body mass throughout the experiment (*Figure 4E*). Accumulated food intake also increased in ghrelin-treated wild-type mice from day 5 of osmotic pump implantation (*Figure 4F*), however the 24 hr daily intake difference between genotypes was not sufficient to conduct pair feeding. We also investigated the effect of prolonged ghrelin treatment on adipose tissue gene expression and found no difference in the expression of thermogenic and oxidative genes in brown fat of ghrelin-treated wild-type and ACC DKI mice (*Figure 4—figure supplement 1B*), indicating that ghrelin's suppressive effect on thermogenesis does not require ACC1 Ser$^{79}$/ACC2 Ser$^{212}$ phosphorylation. Of all the lipogenic genes examined in epididymal white fat, only *Fas* mRNA expression was significantly reduced in ACC DKI tissue (*Figure 4—figure supplement 1B*). However, analysis of adiposity and lean mass by

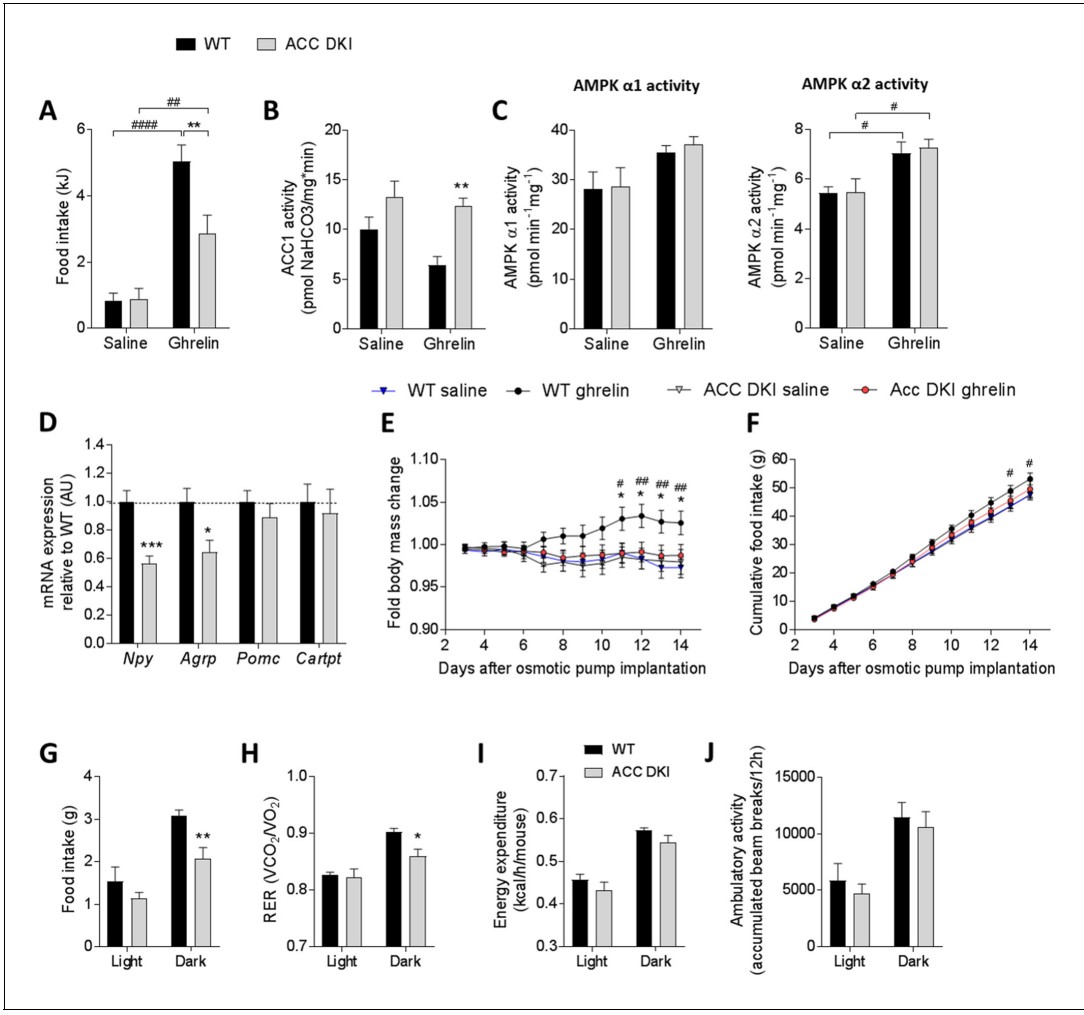

**Figure 4.** Ghrelin-induced food intake and hypothalamic signaling is reduced in ACC DKI mice. (**A**) Food intake (n = 19–20), (**B**) hypothalamic ACC1 activity (n = 4–5) and (**C**), hypothalamic AMPK α1 and AMPKα2 activities (n = 4–5) in wild-type and ACC DKI mice after intraperitoneal injection with saline or ghrelin (1 μg/g body weight). (**D**) Relative mRNA expression of neuropeptides after intraperitoneal injection with ghrelin (1 μg/g body weight) (n = 5–9). (**E**) Change in body mass and (**F**) cumulative food intake during 14 days of saline or ghrelin treatment (30 μg/day/mouse) using osmotic minipumps (n = 8). (**G**) Food intake, (**H**) respiratory exchange ratio, (**I**) energy expenditure and (**J**) ambulatory activity in ghrelin-treated mice on day 14 after osmotic pumps implantation (n = 4–8 mice). (**A**)-(**C**) were analyzed by 2-way ANOVA with Bonferroni post-hoc test with **p<0.01 representing differences between genotypes and p<0.01, p<0.0001 showing differences in food intake or enzyme activity at saline versus ghrelin treatment. (**D**) Data were analyzed by unpaired t-test, two tailed with *p<0.05, ***p<0.001 showing genotype differences in mRNA expression for a given gene. (**E**) and (**F**) were analyzed by 2-way repeated measures ANOVA with Bonferroni post-hoc test with *p<0.05, **p<0.01 showing genotype differences in ghrelin-injected mice and p<0.05, p<0.01 showing differences between body mass or food intake in wild-type mice with saline or ghrelin treatment at a given time point. (**H**)-(**J**) were analyzed by 2-way repeated measures ANOVA with Bonferroni post-hoc test with *p<0.05, **p<0.01 representing genotype differences in metabolic parameters within the dark cycle. All data are presented as mean ± s.e.m.

DOI: https://doi.org/10.7554/eLife.32656.017

The following source data and figure supplements are available for figure 4:

**Source data 1.** Sample size, mean and s.e.m. and statistical calculations are presented.
DOI: https://doi.org/10.7554/eLife.32656.020
**Figure supplement 1.** Effect of long-term ghrelin treatment on adipose tissue gene expression and adiposity in wild type and ACC DKI mice.
DOI: https://doi.org/10.7554/eLife.32656.018
**Figure supplement 1—source data 1.** Sample size, mean and s.e.m.
DOI: https://doi.org/10.7554/eLife.32656.019

NMR or measurements of subcutaneous and epididymal fat pad mass showed no significant differences between genotypes (*Figure 4—figure supplement 1, D–G*), suggesting that the effect of reduced *Fas* expression may have been offset by the increased ACC activity in ACC DKI fat (*Figure 1A*). Furthermore, the difference in body weight was therefore likely due to the small, but accumulated difference in food intake. At day 14 of the treatment, we placed ghrelin-treated wild-type and ACC DKI mice into metabolic cages for 24 hr at 21°C for measurement of metabolic parameters. 12 hr accumulated food intake (*Figure 4G*) and 12 hr average RER (*Figure 4H*) were reduced during the dark cycle in ACC DKI mice, while energy expenditure (*Figure 4I*) and ambulatory activity (*Figure 4J*) were comparable to wild-type. This is similar to the phenotype seen after cold exposure, when plasma ghrelin concentrations are elevated, suggesting that the reduced food intake and preference for lipid utilization in ACC DKI mice may have been a consequence of reduced ghrelin sensitivity.

## Inhibiting ghrelin signaling reduces food intake and RER in wild-type but not ACC DKI mice

To further investigate whether defects in ghrelin signaling may be responsible for the metabolic changes observed in ACC DKI mice during cold exposure, we treated wild-type and ACC DKI mice with saline or the ghrelin receptor (GHSR1a) antagonist [D-Lys$^3$]-GHRP-6 (*Asakawa et al., 2003*) (6.7 µmol/kg), twice daily at the onset of the dark and light phase and measured metabolic parameters by indirect calorimetry. Food intake data for one saline-injected wild type mouse were removed due to a scale malfunction. We found that intraperitoneal [D-Lys$^3$]-GHRP-6 injection reduced 12 hr cumulative food intake in wild-type mice (*Figure 5A and C*) without significant effects on energy expenditure (*Figure 5G*). Furthermore, immediately following [D-Lys$^3$]-GHRP-6 injection, wild-type mice showed a rapid drop in RER that was largely maintained over the following 12 hr period at both, 21°C and 14°C (*Figure 5D and F*). In contrast, administration of [D-Lys$^3$]-GHRP-6 to ACC DKI mice had no significant impact on any of the measured parameters (*Figure 5B, C, E, F and G*), indicating that ghrelin signaling is already inhibited in ACC DKI mice and cannot be further suppressed by the GHSR1 antagonist.

In contrast, treatment with [D-Lys$^3$]-GHRP-6 did not cause the reduction in ambulatory activity observed in ACC DKI mice (*Figure 5H*). The effects on locomotor activity may therefore be independent of the GHSR1 pathway and the ghrelin insensitivity of ACC DKI mice is unlikely to be the primary defect that causes reduced activity.

These results show that the metabolic phenotype of ACC DKI mice with respect to food intake, RER and energy expenditure, but not ambulatory activity, can be reproduced by GHSR1 inhibition and may therefore be a consequence of their ghrelin insensitivity. Resistance to ghrelin receptor signaling may be the main defect responsible for the reduced RER and appetite of ACC DKI mice, as further inhibition at the receptor level with [D-Lys$^3$]-GHRP-6 treatment did not lead to additive effects.

## Food intake in response to leptin or high-fat feeding is not affected in ACC DKI mice

The anorexigenic hormone leptin is known to inhibit hypothalamic AMPK activity (*Dagon et al., 2012*; *Minokoshi et al., 2004*) and has previously been suggested to inhibit food intake through ACC activation (*Gao et al., 2007*). We therefore measured the 24 hr feeding response of wild-type and ACC DKI mice after intraperitoneal injection with 1 µg/g leptin, twice daily at the onset of the dark and light phase. When compared to saline-injected control, leptin lowered food intake to a similar extent in both, wild-type and ACC DKI mice (wild-type 19.4 ± 2.7%, ACC DKI 23.2 ± 3.8%, p=0.44) (*Figure 6A*). Furthermore, there was no detectable difference in the leptin-induced phosphorylation of STAT3 (Tyr$^{705}$) in the hypothalamus (*Figure 6B and C*). These results indicate that AMPK phosphorylation of ACC is redundant for leptin-induced acute signaling and suppression of food intake.

In contrast to plasma ghrelin, which is increased under conditions of energy deficit, leptin correlates with increased adiposity and energy surplus, such as with high-fat feeding. We have previously shown that many of the metabolic differences between wild-type and ACC DKI mice on chow diet are not present in mice fed a high-fat diet for 12 weeks (*Fullerton et al., 2013*). For example,

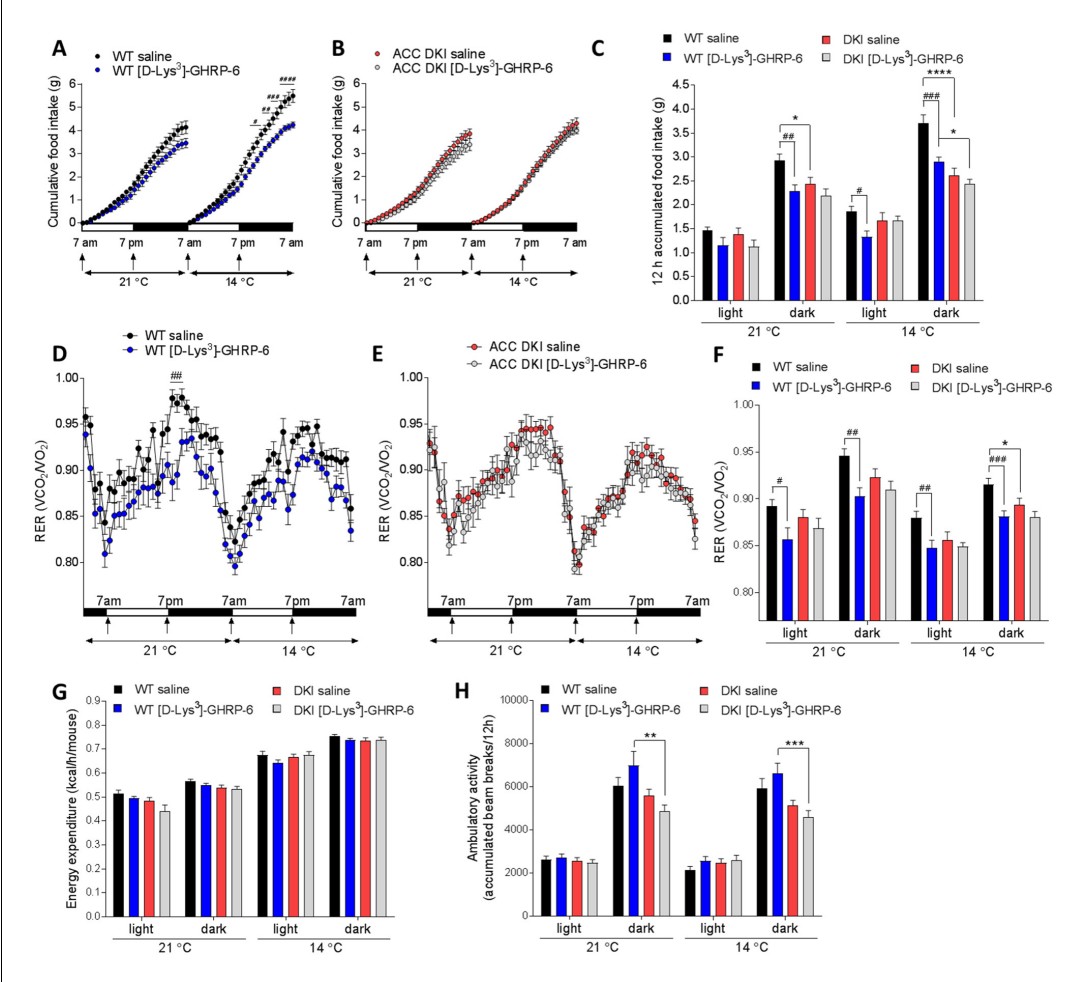

**Figure 5.** Inhibition of ghrelin signaling reduces food intake and respiratory exchange ratio in wild-type but not ACC DKI mice. (A–C) Cumulative food intake, (D–F) respiratory exchange ratio (RER), (G) energy expenditure and (H) ambulatory activity in wild-type and ACC DKI mice at indicated temperatures after daily intraperitoneal injections with [D-Lys3]-GHRP-6 (6.7 μmol/kg, at onset of light and dark cycle). (A, B, D, E) Hourly averages of metabolic parameters of wild-type and ACC DKI mice (n = 12). Black horizontal bars represent the dark period in a 12 hr light/dark cycle (7am-7pm). p<0.05, p<0.01, p<0.001, p<0.0001 represent differences food intake or RER in saline vs [D-Lys3]-GHRP-6 injected mice as determined by 2-way repeated measures ANOVA with Bonferroni post-hoc test. (C, F, G, H) 12 hr averages of metabolic parameters. *p<0.05, **p<0.01, ***p<0.001, ****p<0.0001 represents differences between genotypes within a given light cycle and temperature exposure and p<0.05, p<0.01, p<0.001 are differences between saline and [D-Lys3]-GHRP-6 treatments within the dark or light phase for a given genotype, as determined by 2-way repeated measures ANOVA with Bonferroni post-hoc test. All data are presented as mean ± s.e.m.

DOI: https://doi.org/10.7554/eLife.32656.021

The following source data is available for figure 5:

**Source data 1.** Sample size, mean and s.e.m. and statistical calculations are presented.

DOI: https://doi.org/10.7554/eLife.32656.022

adiposity, RER, insulin sensitivity and hepatic glucose production are similar in high-fat fed wild-type and ACC DKI mice. Food intake has not been examined, but given that high-fat feeding is known to decrease AMPK activity in the hypothalamus and other tissues (*Lindholm et al., 2013*; *Martin et al., 2006*), it is possible that the effect of the DKI mutation on appetite is lost under these conditions.

We initially confirmed that ACC DKI mice fed a high-fat diet at standard animal house temperatures from 6 weeks of age for up to 15 weeks gained weight at a similar rate to wild-type mice (*Figure 6D*). We next measured the average 24 hr food intake over four consecutive days and found no difference between genotypes (*Figure 6E*). Accumulated food intake in response to an acute intraperitoneal injection with ghrelin (1 μg/g body weight) was also similar between wild-type and

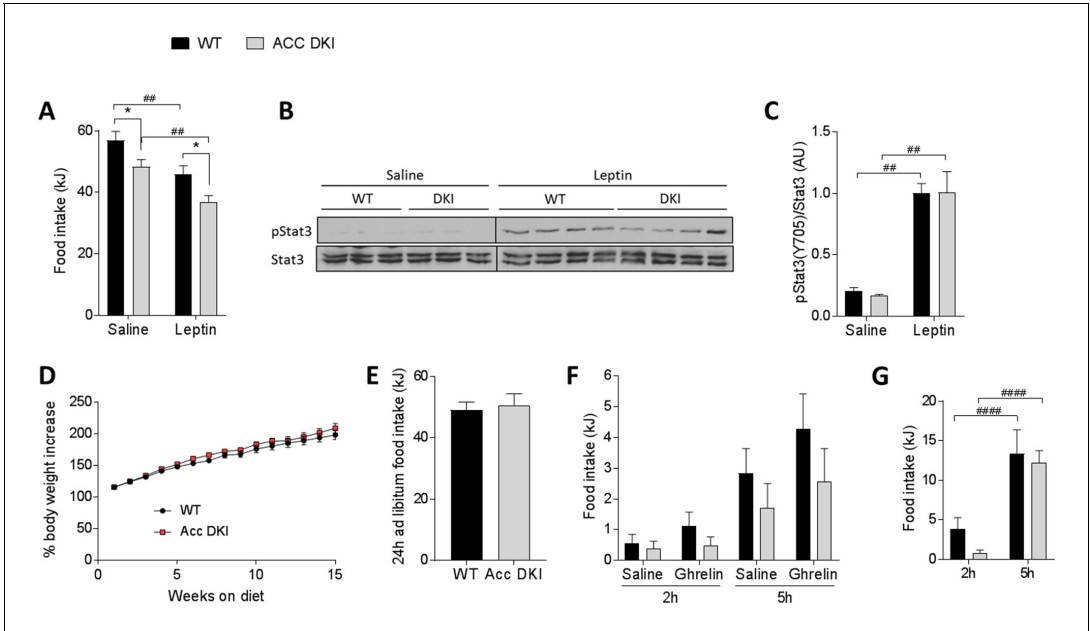

**Figure 6.** Effect of exogenous leptin administration and high-fat feeding on food intake in ACC DKI mice. (**A**) 24 hr food intake after intraperitoneal injection with saline or leptin (1 μg/g body weight) twice daily at the onset of the dark and light cycle (n = 8); *p<0.05 genotype differences, p<0.01 treatment effect as determined by 2-way repeated measures ANOVA with Bonferroni post-hoc test. (**B**) Hypothalamic STAT3 phosphorylation (pTyr[705]) 45 min after intraperitoneal injection with saline or leptin (1 μg/g body weight). Lysates from saline- and leptin-injected mice were run on separate gels, but transferred onto the same membrane for immunoblotting. STAT3 blots were cropped to remove non-specific signals from higher molecular weight proteins in the lysate. (**C**) Quantification of STAT3 phosphorylation normalized to STAT3 total protein signal from the same membrane (n = 3 saline-injected mice, n = 8–9 leptin-injected mice); p<0.01 represents treatment effect as determined by 2-way ANOVA with Bonferroni post-hoc test. (**D**) Body mass increase of wild-type and ACC DKI mice during 15 weeks of high-fat feeding (n = 9); data were analyzed by 2-way repeated measures ANOVA with Bonferroni post-hoc test. (**E**) Average 24 hr food intake measured over four consecutive days (n = 5); data were analyzed by unpaired t-test, two-tailed. (**F**) Accumulated food intake at 2 hr and 5 hr after intraperitoneal injection with saline or ghrelin (1 μg/g body weight) (n = 9) and (**G**) accumulated food intake at 2 hr and 5 hr after overnight (16 hr) fast (n = 9); data were analyzed by 2-way repeated measures ANOVA with Bonferroni post-hoc test; p<0.0001 showing differences in food intake at different time points after re-feeding. All data are presented as mean ± s.e.m.
DOI: https://doi.org/10.7554/eLife.32656.023

The following source data is available for figure 6:

**Source data 1.** Sample size, mean and s.e.m. and statistical calculations are presented.
DOI: https://doi.org/10.7554/eLife.32656.024

**Source data 2.** Western blots are presented for *Figure 6*.
DOI: https://doi.org/10.7554/eLife.32656.025

ACC DKI mice at 2 hr or 5 hr after treatment (*Figure 6F*). It is important to note that regardless of the genotype, ghrelin treatment failed to elicit a statistically significant increase in food intake from saline control and overall intake after ghrelin injection was approximately 5-fold lower when compared to ghrelin-induced intake of chow-fed mice (*Figure 4A*). This is in agreement with previous reports showing that ghrelin is unable to induce a hyperphagic response in diet-induced obesity (*Gardiner et al., 2010*; *Perreault et al., 2004*) and that this effect may be specifically due to ghrelin resistance in NPY/AgRP neurons in the ARC (*Briggs et al., 2010*).

We also investigated the appetite response to an overnight fast in ACC DKI and wild-type mice fed a high-fat diet (*Figure 6G*). There was a tendency for reduced food intake in ACC DKI mice within the first 2 hr after refeeding (p=0.053), however any trend for a genotype difference was lost by 5 hr of refeeding. There was an approximately 5-fold reduction in the overall intake when compared to the response in chow-fed animals (*Figure 3D*), indicating that, similar to the response to ghrelin, high-fat feeding attenuates fasting-induced hyperphagia independently of the DKI mutation.

These data show that the effect of the DKI mutation on appetite are lost with diet-induced obesity and confirm our overall conclusion that AMPK phosphorylation of ACC is of little consequence

for appetite regulation during energy surplus, but is an important signaling step under conditions of energy deficit.

## Discussion

ACC1 Ser[79]/ACC2 Ser[212] phosphorylation is reported in nearly all studies of AMPK physiology and used as a surrogate readout of AMPK activation, leading to the conclusion that this signaling step is required for most functions attributed to AMPK. Using the ACC DKI mouse model we have found that contrary to what was previously assumed, ACC1 Ser[79]/ACC2 Ser[212] phosphorylation is not required under conditions of reduced AMPK activity, such as with increases in thermogenesis and energy expenditure or leptin-induced suppression of food intake. Instead, AMPK phosphorylation of ACC is indispensable for increases in appetite in response to metabolic stress and orexigenic signaling, such as cold exposure, fasting and ghrelin stimulation. It becomes apparent that AMPK is capable of engaging different immediate downstream substrates to mediate these effects.

A number of factors, such as estradiol (*Martínez de Morentin et al., 2014*), nicotine (*Martínez de Morentin et al., 2012*), BMP8B (*Whittle et al., 2012*) and thyroid hormones (*López et al., 2010*) have been shown to inhibit hypothalamic AMPK to reduce sympathetic activity to brown fat. Most of these studies imply AMPK regulation of ACC phosphorylation as an underlying mechanism. However, our results show that the regulation of thermogenic capacity and energy expenditure by AMPK can be mediated independently of ACC Ser[79]/ACC2 Ser[212] phosphorylation and may involve other downstream substrates. These processes may include phosphorylation of transcription factors with subsequent effects on gene expression or reductions of cellular lipid content through substrates other than ACC, such as inhibition of FAS activity.

In contrast, our study confirms that AMPK phosphorylation of ACC is important for increasing appetite. However the importance is limited to effects induced by metabolic stress, such as starvation or short-term cold exposure, whereas the *ad libitum* feeding control can be achieved independently of ACC phosphorylation. Furthermore, our results reveal that the ACC phosphorylation appears to be specifically relevant for the regulation of orexigenic responses. Anorexigenic factors, such as refeeding, leptin or diet-induced obesity, a condition characterized by increased plasma leptin concentrations, are associated with reduced hypothalamic AMPK activity (*Lindholm et al., 2013*; *Martin et al., 2006*; *Minokoshi et al., 2004*). Consequently, the effect of AMPK phosphorylation and inhibition of ACC would be expected to be diminished under these conditions. Suppression of food intake by leptin has previously been suggested to require ACC activation, as administration of the ACC allosteric inhibitor TOFA prevented leptin's anorectic effects (*Gao et al., 2007*). Our results suggest that the ACC activation status in response to leptin can be regulated by other means than ACC1 Ser[79]/ACC2 Ser[212] phosphorylation. Such effects could stem from transcriptional increases in ACC expression or increases in allosteric activators of ACC, such as the TCA cycle intermediate citrate. However, none of these have previously been reported in the literature to occur specifically in response to leptin.

In contrast, the effects of the orexigenic hormone ghrelin on food intake and neuropeptide expression were significantly blunted in ACC DKI mice, showing that of ACC1 Ser[79]/ACC2 Ser[212] phosphorylation by AMPK is an important component of ghrelin signaling and cannot be fully compensated for by other means. However, ghrelin-induced food intake was not completely abolished in ACC DKI mice, suggesting that other pathways must contribute to the ghrelin response. mTOR signaling is a good candidate to mediate this effect, as it has previously been shown to mediate the orexigenic actions of ghrelin (*Lage et al., 2010*; *Stevanovic et al., 2013*; *Zhang et al., 2013*).

Given ghrelin's well accepted role in providing an orexigenic stimulus during fasting and the reduced sensitivity of ACC DKI mice to exogenous ghrelin treatment, it is plausible that the reduced feeding response of ACC DKI mice after a fast may at least in part be due to impaired ghrelin signaling. Whether the ghrelin insensitivity may also be the underlying mechanism for the reduced appetite after cold exposure is less clear. Studies in mice deficient for ghrelin (*Sun et al., 2003*; *Wortley et al., 2004*), the acylated form of ghrelin (ghrelin O-acyl transferase knockout mice) (*Zhao et al., 2010*) or the ghrelin receptor (*Sun et al., 2004*; *Zigman et al., 2005*) have failed to show an effect of ghrelin signaling on spontaneous feeding at standard animal house temperatures and there are conflicting reports with regards to its effect on RER. However, there is evidence that similar to our findings in ACC DKI mice, ghrelin's effect on appetite may be more apparent under

conditions of prolonged negative energy balance. For example, both ghrelin- and ghrelin receptor knockout mice show attenuated feeding in response to repeated overnight fasts (*Abizaid et al., 2006*). Furthermore, the increase in appetite and rebound weight gain after diet-induced weight loss has been shown to be ghrelin dependent (*Briggs et al., 2013*). Under subthermoneutral conditions, mice lacking preproghrelin (which is processed to acyl-ghrelin, desacyl-ghrelin and obestatin) are unable to regulate sleep and body temperature (*Szentirmai et al., 2009*). However, to the best of our knowledge, there are no reports on the effect of ghrelin- or ghrelin receptor deficiency on feeding during prolonged cold exposure. Consistent with our findings in mice, plasma ghrelin has been reported to increase after exposure to cold ambient temperature in rats and humans (*Stengel et al., 2010*; *Tomasik et al., 2005*). In addition, our data show that the use of a ghrelin receptor antagonist reproduces many of the metabolic features of ACC DKI mice at cooler temperatures, such as the reduced food intake, reduced RER and unaltered energy expenditure, indicating that ghrelin receptor signaling may be mediating the increase in appetite during cold stress.

We cannot exclude that in addition to ghrelin signaling, inhibition of ACC phosphorylation may affect other orexigenic pathways. For example, adiponectin (*Kubota et al., 2007*) and endocannabinoids (*Kola et al., 2005*) have also been shown to promote feeding by increasing AMPK activity in the hypothalamus, albeit with additional effects on POMC expression, which we did not observe in our study. Furthermore, it is also possible that signaling initiated by peripheral metabolites may be dysregulated in ACC DKI mice. However, glucose and other metabolites, such as lactate or citrate have all been associated with inhibition of AMPK activity (*Cesquini et al., 2008*; *Cha and Lane, 2009*; *Wolfgang et al., 2007*). These metabolic signals would therefore be expected to lead to a reduction in ACC1 $Ser^{79}$/ACC2 $Ser^{212}$ phosphorylation in wild-type tissue mimicking the effect of the DKI mutation and diminishing any differences in phenotype. We have observed such a redundancy of ACC phosphorylation for appetite control in mice fed a high-fat diet, another condition characterized by low hypothalamic AMPK activity (*Martin et al., 2006*; *Minokoshi et al., 2004*). Obesity is also associated with ghrelin resistance in NPY/AgRP neurons in the ARC leading to reduced ghrelin- and fasting-induced appetite response (*Briggs et al., 2010*; *Gardiner et al., 2010*; *Perreault et al., 2004*). It is possible that the reduced AMPK activity and ACC1 $Ser^{79}$/ACC2 $Ser^{212}$ phosphorylation under these conditions is a contributing factor for the diminished ghrelin response, as AMPK/ACC mediated fatty acid oxidation has previously been shown to be an important activator of NPY/AgRP neuropeptide expression (*Andrews et al., 2008*). However, food intake in both, wild-type and ACC DKI mice was inhibited beyond the values seen in ACC DKI on chow diet, suggesting that the high-fat feeding may overwhelm the effect of ACC phosphorylation by an additional mechanism downstream or independent of the DKI effect.

These results emphasize that the signaling step of ACC1 $Ser^{79}$/ACC2 $Ser^{212}$ phosphorylation is most relevant under conditions of metabolic stress and increased AMPK activity. In *ad libitum* fed mice, reductions in appetite and RER were only apparent at sub-thermoneutral temperatures and were sufficient to prevent an increase in adiposity in these mice despite their propensity for enhanced lipid synthesis in peripheral tissues. Under sufficient metabolic stress the anorexigenic effect of the ACC DKI mutation can dominate over the lipogenic effects in the periphery, but this balance would be expected to shift towards increased fat accumulation in the absence of stress at thermoneutrality. As emphasized previously (*Karp, 2012*), these results show that the environmental temperature is an important factor that needs to be taken into consideration to correctly interpret metabolic experiments in mice and would be of particular importance when investigating the role of stress-activated enzymes, such as AMPK.

In summary, our data identify AMPK phosphorylation of ACC as an important pathway for the regulation of feeding and fuel utilization under metabolic stress. While low ambient temperature and dieting are means to generate negative energy balance, their long-term effectiveness for body weight management is low due to increases in appetite. One year after weight loss 30–35% of patients regain the lost weight and after 5 years 50% will return to their previous body weight and often exceed it (*Wadden et al., 2004*). The weight regain after diet-induced weight loss has been attributed to increased circulating ghrelin concentrations and restored ghrelin receptor sensitivity (*Briggs et al., 2013*). Specific inhibition of ACC1 $Ser^{79}$/ACC2 $Ser^{212}$ phosphorylation may have the potential to reduce ghrelin receptor signaling and alleviate the hunger signal associated with calorie restriction thereby aiding in long-term weight loss management after diet-induced weight loss.

# Materials and methods

## Key resources table

| Reagent type (species) or resource | Designation | Source or reference | Identifiers | Additional information |
|---|---|---|---|---|
| Gene (Mus musculus) | *Acaca* | NA | MGI:108451; Ensembl: ENSMUSG000 00020532 | |
| Gene (Mus musculus) | *Acacb* | NA | MGI:2140940; Ensembl: ENSMUSG000 00042010 | |
| Genetic reagent (Mus Musculus, males) | ACC DKI | doi: 10.1038/nm.3372 | MGI:5780965; MGI:5780967 | C57BL/6-Acaca$^{tm1.1Grst}$, C57BL/6-Acacb$^{tm1.1Grst}$; maintained in B.E. Kemp laboratory |
| Antibody | anti-pSTAT3 (Tyr$^{705}$) (rabbit monoclonal) | Cell Signaling | 9145 | 1:1000 in PBST |
| Antibody | anti-STAT3 (mouse monoclonal) | Cell Signaling | 9139 | 1:3000 in PBST |
| Antibody | anti-pACC (ACC1 Ser$^{79}$/ ACC2 Ser$^{212}$) (rabbit polyclonal) | Cell Signaling | 3661 | 1:1000 in PBS |
| Antibody | anti-AMPK pan$\alpha$ (rabbit monoclonal) | Cell Signaling | 5831 | 1:2000 in PBS |
| Antibody | anti-$\alpha,\beta$ tubulin (rabbit polyclonal) | Cell Signaling | 2148 | 1:5000 in PBS |
| Antibody | anti-AMPK$\alpha$ (Thr$^{172}$) (rabbit monoclonal) | Cell Signaling | 2535 | 1:1000 in PBS |
| Antibody | anti-UCP1 (rabbit polyclonal) | Alpha Diagnostic | UCP11-A | 1:1000 in PBST |
| Antibody | anti-ACC1 (sheep polyclonal) | doi: 10.1038/nm.3372 | | 1:50 in PBS (conjugated to Protein A agarose); against CDEPSPLAKTLELNQ (rat Acc1 (1–15 Cys$^{15}$); G. Hardie laboratory (University of Dundee) |
| Antibody | anti-c-Fos (rabbit polyclonal) | Millipore | ABE457 | 1:1000 in PBS (0.2% Triton X-100, 0.1% BSA, 2% normal swine serum) |
| Antibody | anti-AMPK $\alpha$1 (rabbit polyclonal) | doi: 10.1016/j.chembiol. 2008.10.005. | | 1 $\mu$g in PBS (conjugated to Protein A agarose); against CARHTLDELN PQKSKHQG-COOH (AMPK $\alpha$1 (373–390 Cys$^{373}$); B.E. Kemp laboratory |
| Antibody | anti-AMPK $\alpha$2 (rabbit polyclonal) | doi: 10.1016/j.chembiol. 2008.10.005. | | 1 $\mu$g in PBS (conjugated to Protein A agarose); against CMDDSAMH IPPALKPH-NH$_2$) (AMPK $\alpha$2 (351–366 Cys$^{351}$); B.E. Kemp laboratory |
| Antibody | anti-rabbit immunoglobulin (swine polyclonal, biotinylated) | Dako | E0353 | 1: 500 in PBS (0.2% Triton X-100, 0.1% BSA, 2% normal swine serum) |
| Antibody | anti-rabbit immunoglobulin (goat polyclonal, HRP-conjugated) | Dako | P0448 | 1:3000 in PBST |
| Peptide, recombinant protein | Murine leptin | Lonza (Australia); Peprotech (Rocky Hill, NJ) | 450–31 | 1 $\mu$g/g body weight in saline |
| Peptide, recombinant protein | *n*-octanoylated murine ghrelin | Purar Chemicals | | 1 $\mu$g/g body weight in saline (acute); 30 $\mu$g/mouse at 0.5 $\mu$l/h in saline (osmotic pumps) |
| Commercial assay or kit | Rat/mouse ghrelin ELISA kit | Millipore | EZRGRA-90K | |

*Continued on next page*

*Continued*

| Reagent type (species) or resource | Designation | Source or reference | Identifiers | Additional information |
|---|---|---|---|---|
| Commercial assay or kit | Mouse leptin ELISA kit | Millipore | EZML-82K | |
| Commercial assay or kit | Mouse insulin ELISA kit | Mercodia | 10-1247-01 | |
| Commercial assay or kit | Thyroxine (T4) ELISA Kit | Invitrogen | EIAT4C | |
| Commercial assay or kit | Epinephrine/norepinephrine ELISA kit | Abnova | KA3767 | |
| Commercial assay or kit | NEFA C kit | Wako | 279–75401 | |
| Commercial assay or kit | LabAssay Triglyceride kit | Wako | 290–63701 | |
| Chemical compound, drug | [D-Lys$^3$]-GHRP-6 | Abcam | ab141148 | 6.7 µmol/kg in saline |

## Animals

ACC DKI mice have been generated by intercrossing ACC1 (Ser79Ala) knock-in mice (MGI:5780965) and ACC2 (Ser212Ala) knock-in mice (MGI:5780967) as described previously (*Fullerton et al., 2013*). Male mice were used for all studies and housed in pathogen-free microisolator cages on a 12 hr light-dark cycle. Mice were fed a standard chow diet (9% fat with 13.2 MJ/kg of digestible energy, Barastoc, Ridley Agriproducts, Pakenham, Australia) or placed at 6–7 weeks of age on a high-fat diet (23.5% fat with 17 MJ/kg digestible energy, Specialty Feeds, Glen Forrest, Australia). For all experiments, mice were gender-matched and age-matched within two weeks of age, but otherwise randomized to their respective groups. The St. Vincent's Hospital (Melbourne, Australia) Animal Ethics Committee approved all experimental procedures.

## Enzyme activity assays

ACC activity was measured by $^{14}CO_2$ fixation into acid-stable products with ACC1 protein immunoprecipitated from 1 mg of fat tissue or 0.75 mg of whole hypothalamus tissue using an ACC1 specific antibody as previously described (*Fullerton et al., 2013*). For AMPK activity assays, AMPKα1 and AMPKα2 were immunoprecipitated from 0.75 mg of whole hypothalamus tissue and enzyme activity determined using SAMS peptide in the presence of 200 µM AMP as previously described (*Scott et al., 2008*).

## Adipose tissue lipogenesis and oxidation

For lipogenesis assays, brown and subcutaneous adipose tissue was isolated from mice 1 hr after intraperitoneal injection with 2.5 µCi/g of [$^3$H] acetate (sodium) (Perkin Elmer, Waltham, MA). The lipid fraction was extracted after homogenization in chloroform:methanol (2:1) and radioactivity determined by liquid scintillation counting. For oxidation experiments, brown and subcutaneous fat explants were isolated from mice and transferred to flasks containing essential Krebs-Henseleit buffer (pH 7.4 with 2 mM pyruvate and 1 mM L-carnitine) gassed with 95% $O_2$ and 5% $CO_2$ at 30°C in the presence of 0.2 mM palmitate conjugated to 2% fatty acid-free bovine serum albumin (Bovogen Biologicals, Keilor East, Australia). After 20 min explants were transferred to a similar buffer supplemented with 0.5 µCi/ml of [1-$^{14}$C]palmitate (Perkin Elmer) and tissues incubated for a further 60 min. Medium was removed and acidified with equal volume of 1M acetic acid in an airtight vial. [$^{14}$C]$CO_2$ was trapped in 400 µl benzethonium hydroxide for 60 min and radioactivity measured by liquid scintillation counting. Tissue pieces were washed in ice-cold PBS and lipids extracted by homogenization with chloroform:methanol (2:1) and the radioactivity of the acid soluble intermediates determined as previously described (*Fullerton et al., 2013*). Rates of fatty acid oxidation were determined as a function of both [$^{14}$C]$CO_2$ and incomplete oxidation products.

## Western blotting and quantitative real-time PCR

Tissues were dissected rapidly and snap-frozen in liquid nitrogen and stored at −80°C until further analyses. Expression and phosphorylation of proteins was measured by SDS-PAGE and Western blot using the primary antibodies specific for the following proteins: phosphorylated Tyr$^{705}$ STAT3 (#9145), phosphorylated Ser$^{79}$/Ser$^{212}$ ACC (#3661), phosphorylated Thr$^{172}$ AMPK (#2535), STAT3

(#9139), AMPK panα (#5831), tubulin (#2148) from Cell Signaling Technology (Danvers, MA). The UCP1 antibody (UCP11-A) was from Alpha Diagnostic (Paramus, NJ). Proteins were detected using the ECL method after incubation with horseradish peroxidase (HRP)-conjugated rabbit antibodies (P0448, Daisy, Glostrup, Denmark) as described previously (*Steinberg et al., 2010*). ACC1 and ACC2 were determined using streptavidin-HRP (VWR International, Radnor, PA). For mRNA expression analysis tissues were homogenized using Tri-Reagent (Sigma-Aldrich) and RNA isolated as per manufacturer's instructions. cDNA was generated using the Thermoscript RT-PCR system (Life Technologies, Carlsbad, CA) and analyzed with quantitative Real-time PCR on a Rotorgene 3000 (Corbett Research; Qiagen, Hilden, Germany) using Assay-on-Demand gene expression assays (Life Technologies) according to the manufacturer's recommendations. Assays were normalized using 18S ribosomal RNA and expression calculated using the comparative critical threshold (Ct) method. A list of the TaqMan Gene Expression Assays used in this study is shown in *Supplementary file 1*.

## Immunohistochemistry

Ninety minutes after acute temperature challenge (at 4°C or 30°C) or 60 min after intraperitoneal injection with 1 μg/g ghrelin, mice were anaesthetized and transcardially perfused with PBS followed by fixative (4% paraformaldehyde). Brains were collected and post-fixed in 4% paraformaldehyde and placed in 30% sucrose overnight before coronal sections were taken at 30 μm on a cryostat. After quenching endogenouse peroxidases with 1% $H_2O_2$, sections were blocked with 2% normal goat serum and incubated with rabbit anti-c-Fos antibody (ABE457; Millipore, Billerica, MA) overnight at 1: 1000. After several washes, the sections were incubated for 2 hr with biotinylated swine anti-rabbit antibody (E0353, Daisy) at a dilution of 1:500 at room temperature, then washed and incubated with avidin-biotin complex (Vectastain, Vector laboratories, Burlingame, CA). c-Fos immunoreactivity was visualized after diaminobenzidine (DAB) reaction for 2 min. Images were taken with 100x magnification using a Leica DM 2000 light microscope and relayed with an Olympus DP72 camera. For quantification of c-Fos immunoreactivity, the number of c-Fos positive cells within a constant and defined frame was counted from 3 to 5 consecutive sections per mouse brain using ImageJ software.

## Food intake experiments

For assessment of fasting-induced food intake, food was removed at 1600 hr and returned to the cage the next day at 0900 hr. Food intake was measured over the following 5 hr. For leptin-induced food intake, mice were fed ad libitum throughout the experiment. Mice were injected intraperitoneally twice daily, at the onset of the dark and light phase with leptin (1 μg/g body weight) and 24 hr food intake measured over two consecutive days. Food intake measurements in response to saline injections two days prior to the experiment were used as controls. Recombinant leptin used for intraperitoneal injections was from Lonza (#450–31, Tullamarine, Australia). For acute ghrelin-induced food intake, mice were injected intraperitoneally with saline or ghrelin (1 μg/g body weight) and food intake measured 2 hr after injection. 4 days later mice that received saline were treated with ghrelin and those that received ghrelin were treated with saline and data from both experiments were combined. To determine the plasma ghrelin concentration achieved after acute ghrelin treatment, whole blood was collected 1 hr after intraperitoneal injection with ghrelin (1 μg/g body weight) or with an equivalent volume of saline and processed for analysis by ELISA as described below. For chronic ghrelin treatments, ghrelin (30 μg/day/mouse at 0.5 μl/h for 14 days) or saline was delivered using osmotic minipumps (Alzet, Cupertino, CA) implanted subcutaneously on the dorsal body surface. Food intake and body weight was recorded daily at approximately 1000 hr for 14 days. At day 14 mice were placed into comprehensive mouse metabolic monitoring system for 24 hr for indirect calorimetry. The following day, mice were culled and blood and tissues taken for biochemical analyses. Synthetic *n*-octanoylated mouse ghrelin peptide used for in vivo treatments was supplied by Purar Chemicals (Doncaster, Australia) and purified using low pressure C18 reversed-phase chromatography (0.1% TFA buffer with a 0–60% acetonitrile gradient).

## Metabolic studies

Body composition analysis was performed by nuclear magnetic resonance imaging (Whole Body Composition Analyzer, EchoMRI, Houston, TX). Energy expenditure was measured by indirect

calorimetry in 12–14 week old individually housed mice using the Comprehensive Laboratory Animal Monitoring System (CLAMS, Columbus Instruments, OH). Mice were fed a standard chow diet ad libitum and kept on a 12 hr light/dark cycle at 14°C, 21°C or 28°C with $O_2/CO_2$ consumption and production, food intake and ambulatory activity measured continuously for 72 hr after a 24 hr acclimatization period. At the end of the metabolic measurements, mice were culled and bloods and tissues taken for biochemical analyses. For metabolic studies using ghrelin receptor inhibition, baseline metabolic parameters were measured in mice maintained at 21°C for 24 hr. Mice were subsequently injected intraperitoneally with 6.7 μmol/kg [D-Lys$^3$]-GHRP-6 (ab141148, Abcam, Cambridge, UK) or equivalent volume of saline twice daily at the onset of the light and dark phase and metabolic parameters measured for 24 hr at 21°C followed by 24 hr at 14°C. For plasma hormone measurements, whole blood was collected using the submandibular method into tubes containing $K_3$ EDTA and treated with AEBSF (final concentration of 1 mg/ml). Plasma was acidified with HCl (final concentration of 0.05 N) and hormone concentrations analyzed by ELISA using kits from Millipore for leptin (EZML82K) and acylated ghrelin (EZRGRA90K), Mercodia (Uppsala, Sweden) for insulin (#10-1247-01), Invitrogen (Carlsbad, CA) for thyroxine (EIAT4C) and Abnova (Taipei, Taiwan) for norepinephrine and epinephrine (KA3767). Plasma non-esterified fatty acids (#279–75401) and triglycerides (#290–63701) were measured using colorimetric assays from Wako (Osaka, Japan). For tissue norepinephrine content measurements, brown fat was weighed and homogenized in 0.01 N HCl, 1 mM EDTA, 4 mM $Na_2S_sO_5$ and centrifuged at 13,000 rpm for 10 min at 4°C. Supernatants were analyzed by catecholamine ELISA (KA3767).

## Statistical analyses

All data are presented as mean ± s.e.m. and subjected to statistical analysi using GraphPad Prism 7 software. Statistical significance was determined using two-tailed Student's t-test for single variables (difference between genotypes). Sample size was estimated on previously published studies of our and other's research groups (Claret et al., 2007; Fullerton et al., 2013). For in vivo metabolic studies a minimum of 8 mice per group were analyzed. (1) mRNA expression data, (2) c-Fos immunohistochemistry and (3) ACC1 activities and food intake data comparing a single variable were analyzed by two-tailed, unpaired t-test unless stated otherwise. (1) Age-dependent body weight changes of mice on chow or high-fat diets, (2) daily body weight changes and food intake over time after osmotic pump implantation, (3) age-dependent body composition measurements and fat pad weights, (4) average hourly food intake, RER, energy expenditure and ambulatory activity in metabolic cages, (5) weekly food intake of chow-fed mice and saline-versus ghrelin-induced food intake were analyzed by repeated measures 2-way ANOVA followed by Bonferroni post-hoc test. All remaining data was analyzed by regular 2-way ANOVA with Bonferroni post-hoc test. Significance was accepted at p≤0.05.

## Acknowledgement

This work was supported by grants and a Fellowship (BEK) from the National Health and Medical Research Council (1068813 and 1085460) and the Victorian Government Operational Infrastructure Support Scheme. ZBA was supported by a career development fellowship from the National Health and Medical research council (1084344) and GRS is supported by a Canada Research Chair and the J Bruce Duncan Endowed Chair in Metabolic diseases.

## Additional information

### Funding

| Funder | Grant reference number | Author |
|---|---|---|
| National Health and Medical Research Council | 1068813 and 1085460 | Sandra Galic<br>Gregory R Steinberg<br>Bruce E Kemp |
| Canada Research Chairs | | Gregory R Steinberg |
| J Bruce Duncan Endowed Chair in Metabolic diseases | | Gregory R Steinberg |

| National Health and Medical Research Council | 1084344 | Zane B Andrews |
| Victorian Government Operational Infrastructure Support Scheme | Infrastructure | Bruce E Kemp |

The funders played no role in study design, data collection and interpretation, or the decision to submit the work for publication.

## Author contributions
Sandra Galic, Conceptualization, Data curation, Formal analysis, Funding acquisition, Investigation, Methodology, Project administration, Validation, Writing—original draft; Kim Loh, Conceptualization, Methodology, Writing—review and editing; Lisa Murray-Segal, Methodology, Project administration; Gregory R Steinberg, Conceptualization, Funding acquisition, Supervision, Writing—review and editing; Zane B Andrews, Conceptualization, Investigation, Writing—review and editing; Bruce E Kemp, Conceptualization, Funding acquisition, Resources, Supervision, Writing—review and editing

## Author ORCIDs
Sandra Galic (iD) http://orcid.org/0000-0002-7611-5619
Zane B Andrews (iD) https://orcid.org/0000-0002-9097-7944
Bruce E Kemp (iD) http://orcid.org/0000-0001-6735-5082

## Ethics
Animal experimentation: This study was performed in strict accordance with the approved procedures of the St. Vincent's Hospital (Melbourne, Australia) Animal Ethics Committee (AEC 023/13 and AEC 008/16).

## Decision letter and Author response
Decision letter https://doi.org/10.7554/eLife.32656.029
Author response https://doi.org/10.7554/eLife.32656.030

# Additional files
## Supplementary files
• Supplementary file 1. Table 1: List of TaqMan Gene Expression Assays used for qRT-PCR. Assays were purchased from Applied Biosystems and consist of a pair of unlabeled PCR primers and a TaqMan probe with a FAM dye label.
DOI: https://doi.org/10.7554/eLife.32656.026
• Transparent reporting form
DOI: https://doi.org/10.7554/eLife.32656.027

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
