## [Decision Letter]

Thank you for submitting your article "AMPK signaling to acetyl-CoA carboxylase is required for fasting- and cold-induced appetite but not thermogenesis" for consideration by *eLife*. Your article has been favorably evaluated by Philip Cole (Senior Editor) and two reviewers, one of whom is a member of our Board of Reviewing Editors. The following individual involved in review of your submission has agreed to reveal his identity: David E James (Reviewer #2).

The reviewers have discussed the reviews with one another and the Reviewing Editor has drafted this decision to help you prepare a revised submission.

Summary:

The development of the mouse model with double knockin of mutated AMPK phosphorylation sites in ACC is shown to be useful for assessing the effects of fasting and cold exposure on appetite and energy expenditure. The data support the conclusion that inhibition of ACC phosphorylation leads to reduced appetite in fasting and cold exposure, in spite of maintaining body temperature in the cold. Impaired responsiveness to ghrelin is demonstrated in these double knockin mice as well, supporting your conclusion that ghrelin signaling requires these phosphorylation sites in ACC.

Essential revisions:

There are a few issues we would like you to address in submitting a revised version of your manuscript. First, since you raise potential therapeutic implications, this should be clarified. In reality most obese individuals are in constant positive energy balance and so would targeting this pathway have any utility in the clinic? Use of ghrelin antagonists are already available. Secondly, the interpretation of some of the data is clouded by lack of expected responses in control mice. For example in Figure 4 there was no significant effect of ghrelin on food intake in WT Mice? Also there was not a change in AMPK activity. Also related to the claimed ghrelin resistance, DKI mice had increased circulating ghrelin levels and so why did this not compensate for the block in ghrelin action. In the experiments where ghrelin was injected, what were the circulating levels of ghrelin in both groups of mice?

---

## [Author Response]

Essential revisions:There are a few issues we would like you to address in submitting a revised version of your manuscript. First, since you raise potential therapeutic implications, this should be clarified. In reality most obese individuals are in constant positive energy balance and so would targeting this pathway have any utility in the clinic? Use of ghrelin antagonists are already available.

Our results show that inhibition of ACC Ser79/Ser212 phosphorylation is required to increase appetite in response to a negative energy balance. This becomes clinically relevant when obese individuals are exposed to conditions that confer a negative energy balance, such as occurs with calorie restriction during dieting regimens or increased energy expenditure (cold exposure). Diet-induced weight loss is known to correlate with increased plasma ghrelin concentrations (Cummings DE, et al. (2002), N Eng J Med 346:1623-1630) and restored ghrelin sensitivity in hypothalamic NPY/AgRP neurons (Briggs DI, et al. (2013)). In addition, it has been shown that the rebound weight gain after calorie- restriction is ghrelin dependent (Briggs DI, et al. (2013)) and that reducing circulating ghrelin concentrations is an effective strategy to lower post-dieting increases in food intake and suppress body weight rebound (Chen VP, et al. (2017), PNAS 114: 10960-10965).

Based on our findings we anticipate that inhibition of ACC phosphorylation may reduce weight loss-induced increases in hypothalamic ghrelin sensitivity and help prevent increases in appetite and rebound weight gain after calorie restriction. Therefore, this pathway may be clinically relevant to millions of obese or overweight people trying to prevent rebound weight gain after diet-induced weight loss. We have expanded the last paragraph of the Discussion to clarify potential clinical applications.

Secondly, the interpretation of some of the data is clouded by lack of expected responses in control mice. For example in Figure 4 there was no significant effect of ghrelin on food intake in WT Mice?

There may be a misunderstanding, as Figure 4 shows a statistically significant difference in food intake between saline- and ghrelin-treated wild type mice (0.83 ± 0.23 kJ after saline versus 5.07 ± 0.52 kJ after ghrelin treatment).

Also there was not a change in AMPK activity.

Our original Figure 4 showed a strong tendency for increased AMPK α1 activity with ghrelin treatment, but with great variability between the samples, especially in saline treated controls. We have now investigated the activity of AMPK α2, which has been reported to be the predominant isoform expressed in neurons (Turnley AM, et al. (1999), J Neurochem 72:1707-1716) and better represents the change in AMPK activity in ghrelin-responsive neurons. The new data set is now presented alongside the AMPK α1 activity in Figure 4. The activity of AMPK α2 is approximately 5-fold lower than the activity of AMPK α1.

Also related to the claimed ghrelin resistance, DKI mice had increased circulating ghrelin levels and so why did this not compensate for the block in ghrelin action.

We propose that the defect in ghrelin signalling in the ACC DKI mutant mice is downstream of the ghrelin receptor. Increasing extracellular concentrations of ghrelin may therefore not be able to compensate as the signal cannot be efficiently transduced to downstream effectors. This implies that ACC Ser78/Ser212 phosphorylation is an obligatory signalling component of the ghrelin receptor pathway.

In the experiments where ghrelin was injected, what were the circulating levels of ghrelin in both groups of mice?

We have now measured the plasma ghrelin concentrations in mice 1 h after intraperitoneal injection with 1 μg/g body weight ghrelin or equivalent volume of saline. We have found that this dose resulted in an approximately 3.3-fold increase in plasma ghrelin, when compared to saline-injected controls. Overall, the plasma ghrelin levels achieved after exogenous ghrelin administration reached on average 1.5-fold of the fasting-induced ghrelin concentrations. The new data are presented in Figure 4—figure supplement 1.